# Local Differential Privacy Protection of High-Dimensional Perceptual Data by the Refined Bayes Network

**DOI:** 10.3390/s20092516

**Published:** 2020-04-29

**Authors:** Chunhua Ju, Qiuyang Gu, Gongxing Wu, Shuangzhu Zhang

**Affiliations:** 1Department of Modern Business Research Center, Zhejiang Gongshang University, Hangzhou 310018, China; jch@zjgsu.edu.cn (C.J.); ywwgx@zjgsu.edu.cn (G.W.); zhangshuangzhu0917@126.com (S.Z.); 2School of Management Science & Engineering, Zhejiang Gongshang University, Hangzhou 310018, China; 3School of Business Administration, Zhejiang Gongshang University, Hangzhou 310018, China

**Keywords:** crowd-sensing perception system, perceptual data, high-dimensional data, local differential privacy, the refined Bayes network

## Abstract

Although the Crowd-Sensing perception system brings great data value to people through the release and analysis of high-dimensional perception data, it causes great hidden danger to the privacy of participants in the meantime. Currently, various privacy protection methods based on differential privacy have been proposed, but most of them cannot simultaneously solve the complex attribute association problem between high-dimensional perception data and the privacy threat problems from untrustworthy servers. To address this problem, we put forward a local privacy protection based on Bayes network for high-dimensional perceptual data in this paper. This mechanism realizes the local data protection of the users at the very beginning, eliminates the possibility of other parties directly accessing the user’s original data, and fundamentally protects the user’s data privacy. During this process, after receiving the data of the user’s local privacy protection, the perception server recognizes the dimensional correlation of the high-dimensional data based on the Bayes network, divides the high-dimensional data attribute set into multiple relatively independent low-dimensional attribute sets, and then sequentially synthesizes the new dataset. It can effectively retain the attribute dimension correlation of the original perception data, and ensure that the synthetic dataset and the original dataset have as similar statistical characteristics as possible. To verify its effectiveness, we conduct a multitude of simulation experiments. Results have shown that the synthetic data of this mechanism under the effective local privacy protection has relatively high data utility.

## 1. Introduction

The boom in equipment manufacturing, communication technology, data processing, algorithms, together with the emergence of Internet of Things (IoT), gives rise to the Crowd-Sensing [1,2], a key access to the formation of information value service from the physical world. As shown in Figure 1, various smart devices, which are portable in large space, can realize the perception and digitization of the physical world across time and space. Consequently, large-scale data are acquired for the Crowd-Sensing system [3]. The data are then published to third-party users via sensing servers to perform various analyses, mining and machine learning, ending with providing accurate feedback and decision-making guidance for social production and life [4]. In addition to its large scale, Crowd-Sensing data obtained by immense heterogeneous sensing devices boast the attributes of multidimension or even high-dimensional characteristics in many cases, and, therefore, mining the correlation among the attribute dimensions is vital to the value of Crowd-Sensing data. For example, the correlation analysis of physical features in a patient’s health record helps in the prediction and discovery of potential disease [5], and the correlation analysis of shopping and browsing behaviors of mobile phone users facilitates the personalized recommendation system [6]. Mobile Crowd-Sensing perception is to take the user’s smart mobile device as the basic perception unit, carry out conscious or unconscious collaboration through the mobile internet, realize the distribution of perception and perception data collection, so as to effectively complete the large-scale perception tasks in the fields of urban traffic, society, and environment. With the development of sensors, the modes of Crowd-Sensing perception data tend to be diversified. In addition to sensor data in traditional digital forms, more and more Crowd-Sensing perception data are presented in various forms such as sound, image, and text.

Crowd-Sensing usually contains sensitive information of users, including their environment (such as GPS) and daily behavior (such as step counting). If the sensitive information is misused or released beyond the perceived destination, or cannot be effectively protected in the life cycle of data generation to extinction, it may result in the exposure of perceived user privacy [7,8,9]. Worse still, it might give rise to advertising harassment, economic loss, and even threats to personal safety. Therefore, the protection of Crowd-Sensing data is of particular importance and has been widely addressed by the industry and academia [10]. At present, anonymity-based privacy protection [11] (such as K-anonymity, L-diversity, and T-neighborhood, etc.) and encryption-based privacy protection [12,13] (such as Homomorphic Encryption, secret sharing, security multi-calculations, etc.) are two common methods. However, both anonymity-based and the encryption-based methods fail to meet the demands of strict privacy guarantees and large-scale data processing. Anonymity-based methods often lack strict privacy security guarantees and are thus only suitable for privacy protection of small-scale data [14]. Although the encryption-based methods have better security guarantees, the encryption operation will bring a large computational expense, which makes it difficult to apply to resource-constrained sensing devices [15]. The past decades have witnessed the booming of differential privacy (DP). It has emerged as a standard for privacy protection, due to its rigorous mathematical definition and flexibility in combination. Additionally, due to being light-load, differential privacy is particularly suitable for big data processing and scenarios analysis such as Crowd-Sensing data. However, there are still two major challenges in the application of differential privacy to high-dimensional perceptual data in Crowd-Sensing systems.

The first challenge: non-local privacy protection. Most of the existing privacy protection research focuses on the processing of the collected data, without considering the privacy exposure risks in the data acquisition process. Besides, most research assumes that the data server is a safe place for privacy. In practice, the existing end-to-end encryption ensures that the perceptual data will not be stolen in the communication process, and the centralized differential privacy technology can prevent original perceptual data from third-party thievery via differential and speculative attacks to the published data. However, what is stored in the server is still the unprotected original perceptual data, which is vulnerable to internal attacks [15,16] (such as database leaks and improper operations by server administrators, etc.). Therefore, effective privacy protection should be to realize local privacy protection of the original perception data on the perception device side.

The second challenge: attribute dimension flood. High dimensions and complex correlations [17] among the attribute dimensions make it almost impossible to put protection on every dimension [18]. What is more, the direct privacy protection of high-dimensional data, under the same privacy guarantee, makes utility of the perceived data low and computational expenses larger [19,20]. Therefore, it is a great challenge to protect the privacy of the data while retaining the correlation of the original data.

In response to the above challenges, all kinds of differential privacy protections have been put forward successively, but their application is still less satisfactory. For one thing, some of these protections provide local privacy protection for distributed systems to a certain extent, but they are unqualified for high-dimensional data because of their low utility or high computational complexity. For another, other protections emphasize the centralized privacy protection of high-dimensional data through degrading the dimensionality of high-dimensional data and adopting low-dimensional privacy protection. Unfortunately, these protections fail to provide effective guarantees on local privacy protection for distributed systems, though welcome results have been yielded in so-called “divide and rule”. In order to overcome the difficulties in compatibility between local privacy and high-dimension data in existing privacy protection mechanisms of Crowd-Sensing system, we propose a local privacy protection mechanism for high-dimensional perceptual data based on Bayes network, and the main contributions are as follows.

(1) We propose an aggregation and publication mechanism for high-dimensional perceptual data of local differential privacy. Not only can it provide local privacy guarantees for Crowd-Sensing users, but it can also approximate the statistical characteristics of high-dimensional perception data and publish synthetic data with similar distribution, achieving a good compromise between local differential privacy and high-dimensional data utility.

(2) We also propose an entropy-inspired estimation for the Bayes network construction, which better retains the correlation between attributes and minimizes the calculation amount in the construction process. As a result, we upgrade the efficiency and stability of the algorithm is upgraded to a certain extent.

Finally, we conduct a lot of simulation experiments on the proposed mechanism on multiple real datasets. The experimental results show that the mechanism proposed in this paper retains the attribute correlation of high-dimensional data well, and achieves satisfactory accuracy on the synthetic dataset in both statistical query and analysis tasks.

The remainder of this paper is organized as follows. Section 2 describes related work. Section 3 introduces the system model. In Section 4, we introduce some necessary basic knowledge. Section 5 describes the protection algorithm of local differential privacy on high-dimensional perceptual data. Experimental evaluation results are provided in Section 6. Finally, in Section 7, we conclude this paper.

## 2. Related Work

This paper mainly focuses on local differential privacy protection during the publication of high-dimensional perception data in the Crowd-Sensing system. Therefore, the related work is mainly analyzed and summarized from the aspects of privacy protection for high-dimensional data release and local differential privacy protection research. Differential privacy was designed for protecting a single data record from being speculated via adding an appropriate amount of random noise before the publication. For example, adding sensitivity to the histogram on the data range (the sensitivity in the histogram is the calibrated Laplace noise) is a typical protection before data publication [21,22,23]. As the number of data dimensions grows, the calculation volume of the high-dimensional histogram increases exponentially. Meanwhile, the frequency of most data buckets in the high-dimensional histogram hits zero, which shows great sparsity. In addition, the original protecting noise will result in extremely low Signal to Noise Ratio (SNR), thereby losing the utility of the data. Up to now, studies on safe high-dimensional data publication, in the most cases, have attempted to cut high-dimensional data into multiple low-dimensional data clusters as their first step, taking attribute as their division criteria. PriView [24] constructs *k* marginal distributions of low-dimensional attribute sets, and then estimates the joint distribution of the high-dimensional. However, this method only works based on the assumption that all attributes are independent of each other and attribute pairs are processed equally. Actually, this assumption is not in line with the fact that the attributes in Crowd-Sensing perception systems are associated with each other. Most relevant research sees the correlation between attributes as the criteria for division, such as PrivBayes [19] who adopts the Bayes network to represent the inter-attribute correlation and divides the data by the inter-attribute correlation. However, the method depends too much on sampling the related attribute pairs for index mechanism. When too many attribute pairs are involved, the accuracy of the index mechanism plunges. Accordingly, the Bayes method has been refined by weighting in the literature [25]. Chen et al. [20] introduce dependency graph and joint trees to represent the dimensional association of data. This method calculates the correlation between any two attributes. However, the method is plagued by the complexity of the algorithm, although it is possible to find as much correlation as possible. Markov Chains are also adopted to represent the correlation of data in some other research, but the application on time-related data works better. PrivHD [26] reduces the dimensionality of high-dimensional data via forming Markov nets and segmenting the network to form a joint tree. What is more, it introduces high-pass filtering technology to make differential privacy so as to reduce the search purview of the index mechanism. However, all the methods above are centralized processing, so they are not desirable for the distributed environment of the Crowd-Sensing system.

Local differential privacy [27,28] is a privacy protection intended for distributed environments. Local differential privacy, a conception of differential privacy protection, is a relatively new research area [29,30,31,32,33]. The perturbation mechanism based on the compressed input domain [34], the perturbation mechanism based on information distortion [35], and the local differential privacy implementation of randomized response technology [36] have been proposed in certain literature. The randomized response technology is a major perturbation mechanism for the localized differential privacy. RAPPOR [28] perform local protection by means of randomized response technology, but it is only effective enough for statistical query of low-dimensional data. As the data dimension increases, its communication cost increases exponentially. Kairouz et al. [37] propose the O-RAPPOR method in the case of the unknown value of the attribute variable after the RAPPOR mechanism. O-RAPPOR introduces HashMap and cohort operations to advance the encoding and decoding of RAPPOR. The intention of the introduction is to degrade the impacts of attribute value on the randomized response process. K-RR [38] is another classic method for the release of single-value frequency. Unlike RAPPOR which performs randomized response processing after encoding each value, the K-RR method directly performs randomized responses between multiple values of the variable. Similarly, Kairouz et al. [37] introduce HashMap and grouping operations after K-RR and propose the O-RR method in the case of the unknown value of the attribute variable. Then the O-RR utilizes the perturbation method in K-RR to perform privacy protection treatment, after the process of HashMap. In the case of one-to-many perturbation, the k-Subset method has been put forward in certain research [39], which extends the perturbation output to a form of aggregate. In other words, for a specified single input, it may have multiple output results. In addition, there has been research on the application of local differential privacy on various types of data in recent years, such as image data [40], set-valued data [41], and key-valued data [32]. There is also other research on differential privacy protection in distributed environments. For example, a logistic regression model for differential privacy in the distributed environment has been advanced [42].

## 3. System Model

The Crowd-Sensing system in this paper consists of a large number of sensing users which are connected to a central server. Local privacy protection is carried out after the records of multiple attribute dimensions are perceived and collected, and then the records are sent to the central server. The server collects all locally protected data, estimates and analyzes the statistical distribution of high-dimensional data, and then synthesizes a novel dataset of approximate distribution for third-party users for public query and mining. It must be admitted that the method proposed in this paper does not completely solve the challenge of non-local privacy protection, but it also explores the solution to this challenge to some extent. Here, we mainly focus on data privacy, so we do not consider specific network models.

Assume that there are *N* users in the system, and each user record contains *d* attributes. The aim of data publication is to publish a synthetic dataset of the same size and similar distribution with the original dataset in the central server. Let $X\;=\left\{X^{1},X^{2},\ldots ,X^{N}\right\}$ denote the original dataset, and $X^{i}$ denote the data record of the *i*th user. The attribute set of the dataset is$\;A=\left\{A_{1},A_{2},\ldots ,A_{d}\right\}$, and $x_{j}$ is the value of the corresponding attribute$\;A_{j}$. Thus, the data of a single user is represented as$\;X^{i}=\left\{x_{1}^{i},x_{2}^{i},\ldots ,x_{d}^{i}\right\}$, and $x_{j}^{i}$ is the *j*th attribute of user *i*.
(1)$$P_{X*}\left(A_{1},A_{2},\ldots ,A_{d}\right)\approx P_{X}\left(A_{1},A_{2},\ldots ,A_{d}\right)$$

The range of the attribute is$\;\Omega =\left\{\Omega_{1},\Omega_{2},\ldots ,\Omega_{d}\right\}$, where $\Omega_{j}=\left\{\omega_{j}^{1},\omega_{j}^{2},\ldots ,\omega_{j}^{\left|\Omega_{j}\right|}\right\}$ is the range of the attribute $A_{j}$, $\omega_{j}^{i}$ is the *i*th value of the attribute $A_{j}$, and $\left|\Omega_{j}\right|$ is the range modulo. After receiving all user data, the central server performs a series of data processing and finally releases an approximate synthetic dataset $X^{*}$ with *N* records, which share the range with attribute set *A* of the original dataset $X^{*}$, making the joint probability distribution on the attribute set *A* meet Equation (2).
(2)$$P_{X*}\left(A_{1},A_{2},\ldots ,A_{d}\right)\approx P_{X}\left(A_{1},A_{2},\ldots ,A_{d}\right)$$

$P_{X}\left(A_{1},A_{2},\ldots ,A_{d}\right)≜P_{X}\left(x_{1}=\omega_{1},\ldots ,x_{d}=\omega_{d}\right)$ is the *d*-dimensional joint probability distribution of the attribute set *A* of the original dataset *X*. Here, $x_{i}$ represents the *i*th attribute variable, and $\omega_{i}\in \Omega_{i}$.

## 4. Basic Knowledge

### 4.1. Local Differential Privacy

Differential privacy (DP) [43] is a privacy protection technology that masks real data by adding appropriate random noise to the original data. It has a good mathematical foundation with wide application. Centralized database is the main application of the protection based on differential privacy technology, assuming that the data have been securely acquired and the collectors are trustworthy. However, the database server may not be reliable in terms of privacy security, and therefore local differential privacy (LDP) [38,44] is required. LDP emphasizes that the data perturbation must be performed in the user terminal instead of the central server, so that users can independently process their own sensitive information.

The localized differential privacy protection model fully considers the possibility of data collectors stealing or revealing the privacy of participants (or rather, users) during the data aggregation process. In the localized differential privacy model, each participant (user) performs privacy processing on the data held by him, and then sends the processed data to the central server (i.e., the data collector). The central server performs statistical analysis on the collected data to obtain the analysis results while ensuring that the individual’s private information is not leaked. The definition of local differential privacy is as follows.

Definition 1 (local differential privacy [44]): suppose that *N* users are given, and only one record corresponds to a user. The privacy protection algorithm *M* is given and its domain Dom (*M*) and range Ran (*M*) are defined. If the algorithm *M* yields same output with any two records $X^{i}$ and$\;{\hat{X}}^{i}\;\left(X^{i},{\hat{X}}^{i}\in Dom(M)\right)$, the probability of $X^{*}$ is shown in Equation (3) as follows,
(3)$$P\left(M\left(X^{i}\right)=X^{*}\right)⩽e^{\varepsilon }P\left(M\left({\hat{X}}^{i}\right)=X^{*}\right).$$

Then *M* is qualified for ε—local differential privacy. According to its definition, the output similarity of any two records is great enough to ensure that *M* meets local differential privacy.

The randomized response method [36] is currently the most commonly used technology of local privacy protection, which, in most cases, takes advantage of the uncertainty of the response to protect the original data. Randomized response technology was first adopted in sociological research. While answering private questions, they randomly make decisions between two answers, “Yes” and “No.” Among them, the respondents who give the true answers are with a certain probability *p*, while those who give random answers are with a probability of 1−*p*. In this way, the true response of the respondents cannot be determined, so the privacy of the respondents is protected. What is more, when there is a quantity of respondent responses, true results can be inferred by probability to ensure the effectiveness of the data.

### 4.2. The Bayes Network

The Bayes network is a probabilistic graph model and a directed acyclic graph (DAG) in form, which is often used to deal with the dependencies between variables [20,25]. Assume that *A* is a set of attributes on the dataset *D* and the size of its dimension is *d*. In the Bayes network each attribute in *A* is represented as a node and an edge connecting two nodes indicates the correlation between attributes. If the node $A_{i}$ directly affects the node$\;A_{j}$, a directed arc from $A_{i}$ to $A_{j}$ denotes that the two are causal relationship or unconditionally independent, that is, $A_{i}\to A_{j}$. The core of the Bayes network is conditional probability, which essentially utilizes prior knowledge to establish a correlation of any random variable (attribute). The Bayes network can be regarded as a collection of *d* attribute-parent pairs (AP), where every attribute pair contains an aggregate $\Pi_{i}$ including a node and all its parent nodes. The attribute pair can be represented as$\;\left(A_{i},\Pi_{i}\right)$. Let *N* represent a Bayes network graph and *A* represent the set of all nodes in the network $A=\left(A_{1},A_{2},\ldots ,A_{d}\right)$, then the joint probability distribution of all attributes is expressed in Equation (4) as follows,
(4)$$P\left(A\right)=\overset{d}{\underset{i=1}{\prod }}P\left(A_{i}|\Pi_{i}\right)=P\left(A_{d}|A_{1},\ldots ,A_{d-1}\right)\ldots P\left(A_{2}|A_{1}\right)P\left(A_{1}\right).$$

Figure 2 and Table 1 illustrate the Bayes network. The figure shows the decomposition of five joint nodes into five APs through the Bayes network. In other words, it demonstrates one group of low-dimensional attribute clusters. The joint probability of all nodes $A_{1},A_{2},\ldots ,A_{5}$ in the figure can be calculated by $P\left(A_{1},A_{2},\ldots ,A_{5}\right)=P\left(A_{1}\right)P\left(A_{2}\right)P(A_{3}|A_{1}A_{2})P(A_{4}|A_{1})P(A_{5}|A_{3})$.

## 5. The Protection Algorithm of Local Differential Privacy on High-Dimensional Perceptual Data

Based on the related work, system model and basic knowledge mentioned above, we propose a local differential privacy protection for high-dimensional perceptual data based on the Bayes network in this paper, which qualifies the central server with efficient data publication. Figure 3 presents an overview of this work, which includes three main modules: privacy protection at the local users end sides, dimensionality reduction of high-dimensional perceptual data through Bayes network, and formation of synthetic dataset via sampling and synthesizing. Among them, while the local protection is performed at the local user sides, both the dimensionality reduction of high-dimensional perceptual data and the formation of synthetic dataset are performed on the central server.

### 5.1. Local Privacy Protection

The randomized response technology ensures local privacy protection, but it can only disturb discrete data containing two kinds of values, which is not suitable for multi-valued cases. To deal with multiply-value data, we refer to the variables binarizing in RAPPOR [28], and thereby user data $x_{j}^{i}$ is in the form of binary strings $s_{j}^{i}$ to represent 1. Here, the binary string is constructed mainly according to the value range of the attribute and the position of the attribute value in the value range in this paper. The local user determines the length of the binary string according to the range size $\left|\Omega_{j}\right|$ of the attribute variable $A_{j}$, and each value $\omega_{j}^{i}$ corresponds to a bit in the binary string. Therefore, ${loc}_{i}$ is the bit of the value $\omega_{j}^{i}$. In the data converting, set ${loc}_{i}$ as 1 in the binary string and others as 0, then we can get the unique binary string $s_{j}^{i}$ of the data. What is more, the representation of every attribute value is unique since that the characteristic binary string of every value is independent. As shown in Figure 4, there is a diagram of attribute binarization, and the lower part is the value range of the attribute and the corresponding characteristic binary string.

#### 5.1.1. The Algorithm of Local Privacy Protection

The specific processing of the local differential privacy protection is shown in Algorithm 1, which includes the following three main steps.

Step 1: Binarization. For user *i*, suppose there is an original record $X^{i}=\left\{x_{1}^{i},x_{2}^{i},\ldots ,x_{d}^{i}\right\}$ of *d* attributes, in which $x_{j}^{i}$ denotes the *j*th attribute of the user *i*. The range size is $\left|\Omega_{j}\right|$ in each attribute *A_j_*. The position loc of the value $x_{j}^{i}$ is determined by comparing the original data $x_{j}^{i}$ with the set of attribute value ranges $\Omega_{j}$. The length is $\left|\Omega_{j}\right|$ and loc in string is set as 1.

**Algorithm 1** Data transformation with local differential privacy**Input:** User’s data record {xji|j=1,2,…,d}, attribute set A={A1,A2,…,Ad}, random flipping probability f;**Output****:** Randomized binary string s^i of raw data Xi;1: **for** 1⩽j⩽d
**do**2: Each user *i* transform each attribute *j*th value into a binary string sji;3: Randomly flip each bit of sji to obtain a randomized binary string s^ji;4: **end for**5: Concatenate randomized binary strings for all d attributes to obtain s^i.**Return****:** s^i.

Step 2: Flip the bits randomly. Each bit of the binary string is randomly assigned according to Equation (5) as follows,
(5)$${\hat{s}}_{j}^{i}\left[b\right]=\{\begin{array}{cc} s_{j}^{i}\left[b\right], & \;\mathrm{with}\;\mathrm{probability}\;\mathrm{of}\;1-f\\ 1, & \frac{\;\mathrm{with}\;\mathrm{probability}\;\mathrm{of}\;f}{2}\\ 0, & \frac{\;\mathrm{with}\;\mathrm{probability}\;\mathrm{of}\;f}{2} \end{array}.$$

Step 3: Concatenate the binary string. The binary string ${\hat{s}}_{j}^{i}\left(j=1,2,\ldots ,d\right)$ of every attribute of user *i* is connected and $\overset{d}{\underset{j=1}{\sum }}\left|\Omega_{j}\right|$ bit vector ${\hat{s}}^{i}$ and send it to the central server. In this way, the bit vector is considered to have local privacy protection. 

#### 5.1.2. Privacy Analysis

**Theorem** **1.**
*Assuming that the data record has d attributes and the probability of the randomized response on the user’s local end is f, the local differential privacy level [28] of the users’ end is shown as follows,*



(6)$$\varepsilon =2d\ln\left(\frac{1-\frac{1}{2}f}{\frac{1}{2}f}\right).$$


**Proof.** In the local privacy protection, privacy perturbation of user data is first carried out at the local end, so only the user owns the original data. After sending the data, the possibility of other participants or attackers obtaining the original information is eliminated, and thereby the user’s private information is still under protection. Besides, the central server does not add noise to the collected data after acquiring user data, so the confidence of privacy guarantee in this paper stems mainly from the local processing.Set T as the user’s original binary string and $T^{\prime }$ as the inverted binary string. T and $T^{\prime }$ are two records of two users, respectively. Then the conditional probability ratio $\frac{P(T^{\prime }=T^{*}|T=T_{1})}{P(T^{\prime }=T^{*}|T=T_{2})}$ is related to the privacy level ε and recorded as RR. $T=\left\{t_{1}=1,\ldots ,t_{d}=1,t_{d+1}=0,\ldots \right\}$ can be obtained from the out-of-order original data, based on the fact that only one bit in the binary string of a single attribute in the user’s is 1 (the position of the attribute value is 1). After using the formula, the probability of the unchanged bit value is $1-\frac{1}{2}f$ and the probability of the changed bit value is $\frac{1}{2}f$. According to [28], it can be calculated by
(7)$$\begin{array}{l} RR=\frac{P\left(T^{\prime }=T^{*}|T=T_{1}\right)}{P\left(T^{\prime }=T^{*}|T=T_{2}\right)}⩽\underset{T_{1},T_{2},T^{*}\in Ran\left(M\right)}{max}\frac{P\left(T^{\prime }=T^{*}|T=T_{1}\right)}{P\left(T^{\prime }=T^{*}|T=T_{2}\right)}\\ \;\;\;\;\;⩽\underset{T*\in Ran\left(M\right)}{max}{\left(1-\frac{1}{2}f\right)}^{2\left({t^{\prime }}_{1}+\cdots +{t^{\prime }}_{d}-{t^{\prime }}_{d+1}-\ldots -{t^{\prime }}_{2d}\right)}*{\left(\frac{1}{2}f\right)}^{2\left({t^{\prime }}_{d+1}+\ldots +{t^{\prime }}_{2d}-{t^{\prime }}_{1}-\ldots -{t^{\prime }}_{d}\right)} \end{array}$$Note that only if$\;{t^{\prime }}_{i}=1-t_{i}$, the ratio is the largest, ${\mathrm{RR}}_{max}={\left(\frac{1-\frac{1}{2}f}{\frac{1}{2}f}\right)}^{2d}$, and so $\varepsilon =2dln\left(\frac{1-\frac{1}{2}f}{\frac{1}{2}f}\right)$. □

### 5.2. Synthesis and Publication of High-Dimensional Data after Local Privacy Protection

#### 5.2.1. Conception

The goal of publishing high-dimensional data after local privacy protection is to publish a new dataset that is similar to the original dataset in the statistics (such as probability distribution whose proofs or targets meet Equation (2)). Therefore, there are two direct methods. One is to estimate separately the probability distribution in each dimension and then synthesize high-dimensional data one by one. However, the final synthesized data without considering the correlation among dimensions cannot be subjected to multidimensional joint query and correlation analysis, and thus the value of high-dimensional data is lost. The other is to estimate the probability distribution of all attribute dimensions simultaneously and synthesize a new dataset based on the estimated probability distribution. However, the complete attribute value range will increase exponentially in size as the number of dimensions grows, resulting in great computational complexity and extremely low estimation accuracy. It is evident that the core of keeping high-dimensional data publication under privacy protection lies in choosing an appropriate solution to reduce the dimension and decompose the high-dimensional data into multiple low-dimensional data, to ensure their role of multidimensional joint query and correlation analysis. In this paper, we employ the Bayes network to illustrate the correlation among attribute dimensions in high-dimensional data, and the group features of its probability distribution of multidimensional joint are utilized to estimate the probability distribution of high-dimensional joint. After receiving the processed data of every user, the central server calculates the correlation between attributes by virtue of a joint probability distribution estimation algorithm that is feasible in the low-dimensions data. Then a Bayes network is constructed, and the synthesis and publication of a new dataset is made. Since the perceptual data in this paper are heterogeneous data, mutual information is introduced to measure the correlation between attributes. The core of mutual information calculation is to calculate the joint probability distribution of the two attributes in the locally protected perceptual data. During the construction of Bayes network, we need to solve the mutual information—between a single attribute $A_{j}$ and its parent node set $\Pi_{j}$—and the joint probability distribution between multidimensional attributes. In the following part, we will introduce the estimation algorithm of *m*-dimensional joint probability distribution, and then elaborate the steps of the Bayes network construction.

#### 5.2.2. The Estimation of the Multidimensional Joint Probability Distribution

We mainly extend the Expectation-Maximization algorithm (EM) [45,46] to calculate the joint probability distribution between the multidimensional (such as *m*-dimensional) attributes. According to the self-defined convergence precision δ, the expected value of the probability distribution is obtained through continuous iteration. The specific process is described as follows. Let the *m*-dimensional attribute set be $A=\left\{A_{1},A_{2},\ldots ,A_{m}\right\}$, the index set be C={1,2,…,m}, and the value of attribute $A_{j}$ be $\omega_{j}$. Thereby, the joint probability distribution of *m*-dimensional attribute can be simply denoted as $P\left(\omega_{C}\right)$ or$\;P\left(\omega_{1}\omega_{2}\ldots \omega_{m}\right)$, with *N* users in total. As shown in Algorithm 2, the estimation algorithm of the joint probability distribution of multidimensional data includes the following steps.

Step 1: Parameter initialization. Let the initial joint probability be $P_{0}\left(\omega_{1}\omega_{2}\ldots \omega_{m}\right)=\frac{1}{\left(\prod_{j=1}^{m}\left|\Omega_{j}\right|\right)}$ (Algorithm 2, Line 1).

**Algorithm 2***m*-dimensional joint probability distribution estimation algorithm**Input:** Attributes index set C={1,2,…,m}, randomized binary string s^jt(1⩽j⩽m), flipping probability *f*, convergenceaccuracy δ, attribute set A={A1,A2,…,Am}, attribute domain size Ω={Ω1,Ω2,…,Ωm};**Output:**
*m*-dimensional joint probability distribution P(ωC);1: Initialize P0(ωC)=1(∏j=1m|Ωj|);2: **for** each user i=1,…,N
**do**3: **for** each attribute j∈C
**do**4:  Compute P(s^ji|ωj)=∏b=1|Ωj|(f2)sji[b](1−f2)1−sji|b|;5: **end for**6: Compute P(s^Ci|ωC)=∏j=1mP(s^ji|ωj);7: **end for**8: Set t=1;9: **repeat**10: **for** each user i=1,…,N
**do**11:  Compute Pt(ωC|s^ci)=Pt−1(ωC)P(s^Ci|ωC)∑ωCPt−1(ωC)P(s^Ci|ωC)(ωC∈Ω1×Ω2×…×Ωm)12: **end for**13: Compute Pt(ωC)=1N∑i=1NP(ωC|s^Ci);14: t=t+1;15: until maxωCPt(ωC)−maxωCPt−1(ωC)⩽δ;**Return**: P(ωC)=Pt(ωC).

Step 2: Conditional probability calculation. Calculate the conditional probability of *m*-dimensional data of each user, i.e., $P({\hat{s}}_{1}^{i}{\hat{s}}_{2}^{i}\ldots {\hat{s}}_{m}^{i}|\omega_{1}\omega_{2}\ldots \omega_{m})$. As the meaning of each bit in the user’s binary string is different and the bit flips are independent of each other, it is believed that the conditional probability of *m*-dimensional attribute joint is the product of the conditional probability of each bit, that is, $P({\hat{s}}_{1}^{i}{\hat{s}}_{2}^{i}\ldots {\hat{s}}_{m}^{i}|\omega_{1}\omega_{2}\ldots \omega_{m})=\overset{\left|\Omega \right|}{\underset{b=1}{\prod }}{\left(\frac{f}{2}\right)}^{s_{C}\left[b\right]}{\left(1-\frac{f}{2}\right)}^{1-s_{C}\left[b\right]}$(Algorithm 2, Lines 2–7).

Step 3: Expectation-maximization estimation. The number of initial iterations is t=1 (Algorithm 2 Line 8). The iterative process of expectation-maximization estimation involves the following two steps.

Step E: The calculation of posterior probability. Given the conditional probability of the binary strings of every user, the Bayes probability can be calculated by
(8)$$P_{t}\left(\omega_{1}\omega_{2}\ldots \omega_{m}|{\hat{s}}_{1}^{i}{\hat{s}}_{2}^{i}\ldots {\hat{s}}_{m}^{i}\right)=\frac{P_{t-1}\left(\omega_{1}\omega_{2}\ldots \omega_{m}\right)P\left({\hat{s}}_{1}^{i}{\hat{s}}_{2}^{i}\ldots {\hat{s}}_{m}^{i}|\omega_{1}\omega_{2}\ldots \omega_{m}\right)}{\sum_{\omega_{1}}\ldots \sum_{\omega_{m}}P_{t-1}\left(\omega_{1}\omega_{2}\ldots \omega_{m}\right)P\left({\hat{s}}_{1}^{i}{\hat{s}}_{2}^{i}\ldots {\hat{s}}_{m}^{i}|\omega_{1}\omega_{2}\ldots \omega_{m}\right)}.$$

Here, $P_{t-1}\left(\omega_{1}\omega_{2}\ldots \omega_{m}\right)$ is the results after t=1 iterations (Algorithm 2, Lines 10–12).

Step M: Iteratively update the parameter$\;P_{t}\left(\omega_{C}\right)$. Replace the prior probabilities of the previous round with the average $P_{t}\left(\omega_{C}\right)=\frac{1}{N}\overset{N}{\underset{i=1}{\sum }}P(\omega_{C}|{\hat{s}}_{C}^{i})$ of the posterior probabilities of *N* users to generate a new *k*-dimensional joint probability distribution (Algorithm 2, Line 13), and then return to Step E.

The steps above keep iterating until the difference between the two final joint probability is less than the convergence precision δ, namely, $max_{\omega_{C}}P_{t}\left(\omega_{C}\right)-max_{\omega_{C}}P_{t-1}\left(\omega_{C}\right)⩽\delta$. Here, δ is defined according to the accuracy requirements (Algorithm 2, Line 15).

Generally speaking, if the initial value is a proper selection, the estimation of multidimensional joint probability distribution based on the EM can converge to a just estimated value after a certain number of iterations. However, as the number of dimension *m* increases, the size of the state space after multidimensional combination is $\overset{m}{\underset{j=1}{\prod }}\left|\Omega_{j}\right|$, which is inclined to exponentially grow. As a result, the complexity of the algorithm increases sharply. Moreover, with the increase of the state space, the actual values of many states disappear (that is, the quality of being sparse). However, the EM algorithm may still estimate the probability distribution of these sparse states, which will bring great estimation errors, and thus ultimately show great loss of utility.

#### 5.2.3. Bayes Network Construction

After we obtain the joint probability distribution of arbitrary *m*-dimensional attributes, we can work out the mutual information between the *m*-dimensional attributes subsequently. Generally, the larger the value I is in the mutual information, the more relevant the two attributes are. Suppose we want to build a Bayes network *N* with a maximum number of in-degree *k* (that is, the maximum number of parent nodes of each node is *k*) based on the dataset *D*. Let each attribute in $A=\left\{A_{1},A_{2},\ldots ,A_{d}\right\}$ denote a node in the Bayes network. The network is constructed by collecting nodes from the attribute set one by one. Algorithm 3 below expounds the construction process of the Bayes network.

**Algorithm 3** Bayes network construction algorithm**Input:** Dataset *D*, maximal degree of Bayes network *k*, attribute set *A*;**Output:** Bayes network N;1: Initialize N=0, S=0;2: Randomly pick a node from A as X1 and add it into *S*, add (X1,0) into *N*;3: **for**
i=2 to *d*
**do**4: For ∀X∈A/S and ∀Π∈CSk, add (X,Π) into Ω and compute I(X,Π);5: Choose the attribute-parent pairs (Xi,Πi) with the maximal *I*;6: Add Xi into *S* and add (Xi,Πi) into *N*;7: **end for****Return:** N={(X1,0),(X2,Π2),…,(Xd,Πd)}.

Assume that the set *S* holds the existing attributes, and the initially set is S=0 and *k* denotes the maximum number of parent nodes (Algorithm 3, Line 1). Firstly, we randomly select an attribute from the *d* attributes as the initial node $X_{1}$ for the Bayes network, and then set its parent node set to empty, that is, Π1=0 (Algorithm 3, Line 2). At the same time, $X_{1}$ will be added to the set *S* and (X1,0) will be added to *N*. Secondly, *k* nodes in *S* are selected to obtain a $C_{\left|S\right|}^{k}$ set of all possible parent nodes (when there are less than *k* nodes in *S*, the whole is regarded as one set of parent nodes on the purpose to ensure that the number of parent nodes does not exceed *k*). The parent node sets combined with all the nodes in *A*/*S* give rise to AP (X,Π) which will be stored in Ω. After that, we calculate the mutual information *I* of all AP in Ω (Algorithm 3, Line 4, the detailed calculation of the mutual information will be elaborated later on). Then, the attribute pair with the largest *I* is picked out and added to the Bayes network. Meanwhile, the attribute point is moved to the set *S* (Algorithm 3, Lines 5–6). These steps are repeated before S={1,2,…,d}, and then the Bayes network construction is completed. The correlation calculation mentioned in the fourth line of Algorithm 3 is shown as follows,
(9)$$I\left(X,\Pi \right)=\underset{x\in \mathrm{Ran}\left(X\right)}{\sum }\underset{y\in \mathrm{Ran}\left(\Pi \right)}{\sum }p\left(x,y\right)\log\frac{p\left(x,y\right)}{p\left(x\right)p\left(y\right)},$$
where Ran(X) and Ran(Π) represent the range of the attribute node *X* and the attribute set *Π*, respectively. p(x,y) are joint probabilities when the value (X,Π) is (x,y), which can be figured out by Algorithm 2 in Section 5.2.1. Meanwhile, the attribute dimension m=2, and p(x) as well as p(y) indicate the prior probability when *X* and *Π* take the values *x* and *y*, respectively. p(x) and p(y) can be obtained directly from the joint probability p(x,y) according to the relationship between the joint probability and the edge probability.

#### 5.2.4. The Refined Bayes Network

The Bayes network construction above can decompose high-dimensional attributes and effectively reduce the computing load and improve the utilization of local data publication. However, there are still two weaknesses after observation. For one, the construction of Bayes network is of great uncertainty in every calculation since the initial node is randomly selected in Algorithm 3 (Line 2). Attribute nodes are selected indefinitely and whereby big deviation could be made in the approximate joint probability distribution of the Bayes network. For another, the selection of $I_{\max}$ in Algorithm 3 requires the calculation of mutual information of every attribute pair in Ω. However, in the iteration process, only one node is picked out from V at a time and the weaker attribute pairs will repeat its occurrence in the subsequent calculation of mutual information. It wastes memory and increases the calculation load. In order to overcome these two shortcomings, we propose a better construction algorithm of Bayes network, as shown in Algorithm 4. Please note that the mutual information in Equation (8) can be rewritten into information entropy as follows,
(10)I(X,Π)=H(X)+H(Π)−H(X,Π).

**Algorithm 4** An improved algorithm for Bayes network construction**Input:** Dataset *D*, degree of Bayes network *k*, attribute set *A*;**Output:** Bayes network *N*;1: Initialize N=0, S=0 and V=A;2: Compute the entropy *H* for each attribute in *A*, and choose the attribute with the maximal entropy as X1 add it into S and (X1,0) into N;3: for i=2 to *d* do4: Ω=0;5:  **if** |S|>k
**then**6:   For ∀X∈A/S and ∀{Π∈CSk|Xpick∈Π}, add (X,Π) and (Xj,Πj) into Ω, then cor I(X,Π);7:  **else**8:   For ∀X∈A/S and ∀Π∈CSk,add (X,Π) into Ω, then compute I(X,Π);9:  **end if**10: Choose the attribute-parent pair with the maximal *I* and denote as (Xpick,Πpick);11: Choose the attribute-parent pair with the maximal value of I′(Xj,Πj)(Xj≠Xpick ) and denote as (Xpick ,Πpick );12: Add Xpick  into *S* and (Xpick,Πpick) into *N*;13: **end for****Return:** N={(X1,0),(X2,Π2),…,(Xd,Πd)}


Here, H(x) is the information entropy and shows the uncertainty of a variable. The larger the entropy value is, the greater the uncertainty of the node. H(X,Π) stands for the cross-entropy between the variables *X* and *Π*. Inspired by the formula, the attribute X with the larger information entropy value H(X) is more likely to be picked, when searching for the attribute pair with the largest mutual information value *I*. Therefore, we take the attribute with the largest information entropy as the initial node to construct the Bayes network. In this way, the correlation between attributes in the constructed Bayes network is probabilistically increased with better accuracy of joint query of high-dimensional data. In brief, the initial node should be selected based on entropy value. For details, see the second line in Algorithm 4. The calculation of information entropy is as follows,
(11)$$H\left(x\right)=-\sum p\left(x_{i}\right)\log p\left(x_{i}\right),\;i=1,2,\ldots ,\left|\Omega \right|.\#$$

Here, xi is the *i*th possible value of the attribute *x*, while p(xi) is the edge probability of xi. With regard to the redundant calculation of the attribute mutual information, we modify Lines 4–6 of Algorithm 3 to reduce the number of attribute pairs in Ω. In this way, we can reduce the calculation load. Firstly, when |S|>k and the attribute pairs in Ω is bigger than 1, the mutual information *I* of all attribute pairs is calculated. The attribute pair (Xpick,Πpick) that boasts the largest mutual information and the attributes (Xj,Πj) with the largest mutual information *I* in the remaining attributes of Xj≠Xpick are selected (Lines 9–10 in Algorithm 4). Xpick  is the nodes to be added to the Bayes network. Xpick  and (Xpick ,Πpick) are added to *S* and *N*, respectively (Line 11 in Algorithm 4). Then in the next round of iteration, only the composition attribute pairs made up of ∀X∈A\S and *Π* from Xpick  in the previous round of iteration are stored in *Π*, and the selected (Xj,Πj) in the previous round of iteration is added to *Π* (Algorithm 4 Line 5). Eventually, mutual information is re-calculated and nodes are re-selected, and calculation ends when V = Ø. The whole improvement details are shown in Algorithm 4.

### 5.3. Synthesis of Dataset

From Equation (3), we can independently sample and generate numbers on each attribute according to the conditional probability distribution of the Bayes network, thereby synthesizing a new dataset. Specific steps are as follows.

Step 1. The node (attribute) whose parent node set is 0 in the constructed Bayes network is regarded as the initial sampling nodes $X_{1}$. The node data are sampled according to the edge probability distribution calculated in Section 5.2.1. The number of sampling attribute data is recorded as *N*.

Step 2. From the un-sampled nodes, randomly select a node whose parent node set *Π_i_* has been sampled as the sampling node for this round. Calculate its conditional probability distribution $P(X_{i}|\Pi_{i})$ based on the joint probability distribution in Section 5.2.1. The conditional probability distribution is regarded as the basis for node sampling.

We repeat Step 2 until all attribute nodes are sampled. The sampled data of all nodes constitutes a new *N* × *d* synthetic dataset. The new synthetic dataset to a certain extent holds similar performance in the statistical probability distribution of the original dataset. Since the calculation above is on the processed user data after local privacy protection, the algorithm process as a whole still guarantees local privacy of the Crowd-Sensing users.

## 6. Experimental Evaluation

In this section, the mechanism proposed in the paper is simulated on real datasets. The accuracy is evaluated and analyzed in three aspects, namely Bayes network construction, the multidimensional probability distribution of the synthetic dataset, and classification task of the synthetic dataset.

### 6.1. Experiment Setup

#### 6.1.1. Dataset

In the simulation experiments, we used three real datasets: (1) NLTCS, a dataset of an American nursing survey center, which records the daily activities of 21,574 disabled people at different period (NLTCS is a data set often used to verify the feasibility of local differential privacy algorithm.); (2) Adult, the partial data of 45,222 USA residents from the census 1994 (Adult is a data set often used to verify the feasibility of local differential privacy algorithm.); and (3) TPC-E, a dataset from an online transaction program developed by TPC, recording 40,000 pieces of data in transactions, transaction types, security, and security status (TPC-E is a data set often used to verify the feasibility of local differential privacy algorithm.). In the experiment, for the sake of simplicity, the non-binary datasets Adult and TPC-E were sampled. The attribute value ranges were synthesized and compressed. The detailed information of the three processed datasets are shown in Table 2.

#### 6.1.2. Experiment Methods

All simulation experiments adopted Python 2.7 and the experiment hardware included Intel i5-3470, CPU of 3.20 GHz, memory of 8 GB, and Windows 10. The publication of Crowd-Sensing data was simulated in the following steps. Firstly, the user node reads data in turn from the dataset and a privacy-protected bit string appears after local privacy protection. Then, after sending these bit strings to the central server for learning, a Bayes network model is built up based on Bayes network sampling and synthesis, and finally releasing a new dataset for arbitrary query.

Then, these bit strings are sent to the central server for learning, constructing a Bayes network model, based on Bayes network sampling and synthesis, and finally releasing a new dataset for arbitrary query.

#### 6.1.3. Experiment Parameters

During the simulation, the flip probability *f* of all datasets during local privacy protection ranged from 0.1 to 0.9. In the construction of the Bayes network with the binary dataset NLTCS, the maximum in-degree *k* had four values: 1, 2, 3, and 4. When constructing the Bayes network on a non-binary dataset, the maximum in-degree *k* takes into account two values: 1 and 2.

#### 6.1.4. Evaluation Indicators

The purpose of the experiment was to evaluate the utility of the synthesized data, which has been published under local privacy protection. The utility was mainly evaluated from three aspects. Firstly, find the differences in correlation identification in the Bayes between the synthetic dataset and the original. The correlation identification gap tells the correlation loss in high-dimensional data under local privacy protection. Secondly, compare the mean squared deviation between the edge probability distributions of the synthetic dataset and the original to evaluate the accuracy of edge probabilities on multidimensional attributes in the synthetic dataset. Then, the reliability of statistical query can be measured based on the accuracy. Thirdly, identify the differences in data analysis between the synthetic dataset and the original dataset, such as SVM classification (SVM classification is a classical binary classification algorithm, mainly used to solve the problem of data classification in the field of pattern recognition.). Then the overall utility of high-dimensional data under local privacy protection is evaluated.

### 6.2. Results

#### 6.2.1. Bayes Network Construction and Attribute Correlation Accuracy

In this section, we conduct experiments to study the influence of the values of *k* and *f* on the attribute correlation identification accuracy and entropy-inspired initial node on the construction of Bayes network. In this paper, we employed the Bayes network to model the correlation between attribute dimensions of high-dimensional data. Nodes in the Bayes network represent attribute dimensions. The maximum in-degree *k* of a node in the Bayes network directly affects the number of its parent nodes, and thus affects the calculation of the related attribute pairs. In addition, the flip probability *f* determines the perturbation probability of the users’ data transmitting under local privacy protection. Then *f* shapes the accuracy of the Bayes network construction.

Figure 5 shows the sum of the mutual information $I_{\mathrm{sum}}=\overset{d}{\underset{i=1}{\sum }}I\left(A_{i},\Pi_{i}\right)$ of all attribute pairs in the Bayes network constructed with different *k* and *f.* Mutual information measures the correlation between attributes. The larger the mutual information is, the higher the attribute correlation is. It means that the mutual information $I_{\mathrm{sum}}$ to a certain extent tells how much correlation between attribute dimensions in high-dimensional data has been lost. It can be concluded from Figure 5 that as the value of *k* increases, the $I_{\mathrm{sum}}$ of each dataset increases. In other words, the larger the value of *k* is, the closer the constructed Bayes network is to the full probability distribution Pr[A] of the dataset. However, Figure 5 also demonstrates that after *k* reaching a certain value, the growth of $I_{\mathrm{sum}}$ gets much smaller. This implies that the growth of *k* is no longer adequate in mining the correlation between attributes. In other words, the work of picking attribute pairs with correlation has been completed. Besides, in different datasets, after *f* becomes higher than a certain point, the recognition accuracy of the attribute correlation in the Bayes network almost cannot be altered by the value of *k*. However, as *f* changes, the development of $I_{\mathrm{sum}}$ is different. This is because the calculation of mutual information is related to both the edge probability distribution and joint probability distribution of attributes. With different flip probabilities, the mutual information sum $I_{\mathrm{sum}}$ of all attribute APs will be different even if the Bayes network remains the same, but whether it increases or decreases depends on the data.

Figure 6 compares $I_{\mathrm{sum}}$s of all attributes in the Bayes networks constructed by random selection and entropy-inspired initial nodes, where k=2. It can be seen from the figure that the $I_{\mathrm{sum}}$ as a whole is higher than that of the attributes in the Bayes network constructed by random selection of the initial nodes. This indicates that the entropy-inspired initial nodes better maintain the correlation between high-dimensional attributes than the random selection. In this way, the accuracy of the joint query on the synthetic dataset is ensured. As for a single dataset like the binary dataset NLTCS, the difference between random selection and selection based on information entropy is slight. This is because the sparseness of the binary attribute distribution is low but the correlation between attributes is strong enough. In terms of non-binary dataset, Adult, and TPC-E, the entropy-inspired selection generally has much higher mutual information than the random when the *f* is small and the difference of the two selection is small with large *f* value. This is because when *f* is small, the accuracy of the probability distribution estimation is high, and thus the entropy-inspired method has significant advantages. However, as *f* increases, the estimation deviation in joint probability distribution grows. Then selection in the Bayes network randomizes, and, therefore, the entropy-based one loses its huge advantage.

Figure 7 shows the accuracy of edge connections in the refined Bayes networks with different *f*, compared with the accuracy in the Bayes network constructed with the original raw data. Edges between two attributes in the Bayes network helps in the identification of attributes correlation. Whether there are edges between two attributes in the Bayes network also intuitively reflects the judgment of related attributes. The accuracy of the identification of attribute correlation by the mechanism in this paper can be effectively reflected by comparing the recognition accuracy of the related attributes identified by the model with the original dataset after the privacy protection. It is worth mentioning that because the Bayes network constructed by the original data at different node degrees *k* is also different, there is no direct comparison between different *k*, so here we only look at the impact of *f* on the accuracy of construction. From the experimental results in Figure 7, it can be seen that given different node degrees *k*, as *f* increases, the accuracy of correlation identification between attributes decreases overall. This is because the increased degree of privacy protection will cause a certain degree of loss in the accuracy of the constructed Bayes network.

#### 6.2.2. The Accuracy of Statistical Query

This section is an experiment-based evaluation on the statistical query accuracy of synthetic datasets. Given *a*-dimensional attributes, the experiment compares the joint probability distribution of the synthetic dataset (obtained directly from the synthesized dataset) and the distribution of original dataset to work out the deviation in distribution, and thus the accuracy of statistical query is evaluated. $Q_{a}$ includes all the *a*-dimensional attribute unions. Deviation is measured by the average variation distance, which was adopted in the literature [19,20,47]. The definition is shown as follows,
(12)$$\mathrm{AVD}\left(Q_{a},{\hat{Q}}_{a}\right)=\frac{1}{2}\underset{\omega \subset \Omega }{\sum }\left|Q_{a}\left(\omega \right)-{\hat{Q}}_{a}\left(\omega \right)\right|\;\left|\Omega \right|=16,$$
where, Ω is the range of *a*-dimensional attribute unions, $Q_{a}$ is the joint probability obtained from the real dataset, and ${\hat{Q}}_{a}$ is the joint probability from the synthetic dataset. In addition, the KL divergence is introduced for measurement, which was used in the literature [48].

Figure 8 first shows the comparison between the original joint probability distribution and the joint probability distribution estimations of the NLTCS, Adult, and TPC-E on the different range sizes (|Ω| is 8, 16, 32, respectively), and AVD of the corresponding distribution differences. When the *f* is smaller than 0.1, that is, when the privacy protection is relatively weak, it can be seen that in different value ranges, the joint probability distribution deviations of both the synthetic dataset and the original dataset are small. When the *f* is more than 0.9, that is, when the privacy protection is strong enough, the joint probability distribution on the synthetic dataset still roughly indicates the original distribution in the case of a small range |Ω|=8. The AVD is low, but the effectiveness is still acceptable. In the case of a large range (|Ω|=16 or 32), the deviation of the joint probability distribution estimation of the synthetic dataset is larger, and the corresponding AVD value is also bigger than before. When *f* is moderate, like 0.5, the privacy protection is moderate. At this time, the joint probability distribution of the synthetic dataset indicates the original distribution well in different value ranges, and the AVD value is also small, which reflects a better data utility.

Figure 9 demonstrates the querying deviation of the *a*-dimensional joint probability distribution between the synthesized dataset and the original dataset on the NLTCS, Adult, and TPC-E3, with the node in-degree k=2 in the Bayes network construction and different *f*. Figure 9a–c shows the deviation based on the AVD (average variation distance), while Figure 9d–f demonstrates the changes in the corresponding KL divergence. Here, we mainly compare 2-5-dimensional joint probability, with $Q_{a}$ representing the a-dimensional joint probabilities. The experimental results indicate that, on the whole, for every $Q_{a}$, the average deviation and the KL divergence increase as the value *f* grows. When *f* is large, the average deviation and the KL divergence will surge. This is because larger *f* represents higher degree of user privacy protection. The probability of sending fuzzy data is greater and the distribution deviation from the original data is larger, which implies that there should be a compromise between privacy protection and data utility. Meanwhile, with the same degree of privacy protection *f*, as the query dimension *a* increases from 2 to 5, the corresponding average deviation and KL divergence of $Q_{a}$ will soar. This is because as the query dimension increases, the state space corresponding to the multidimensional combination becomes increasingly sparse, the error of the multidimensional joint probability distribution estimation will also increase, and the data utility is obviously lost. This also explains the reason why it is necessary to reduce the dimensionality of the high-dimensional data in order to ensure that the local privacy protection can still recover better data utility, which is consistent with the conclusions in this paper.

#### 6.2.3. Classification Accuracy

This section is an experiment-based evaluation on the accuracy of multidimensional data analysis on synthetic datasets. We trained multiple SVM classifiers on three original datasets and three synthetic datasets on each dataset, and then obtained and compared their average test accuracies. In this experiment, 80% of the records in each dataset were taken as the training set and the other 20% of the records were used as the test set. Each binary attribute of the dataset was used successively as a label for classification before the training of multiple classifiers. Every classification task was simulated five times and then the average classification accuracy was calculated.

Figure 10 depicts the average accuracy of SVM classification on NLTCS, Adult, and TPC-E by our method. Figure 11 depicts the average accuracy of SVM classification on NLTCS, Adult, and TPC-E by MeanEST algorithm (MeanEST algorithm is a commonly used mean estimation local difference privacy algorithm in academia.) [49]. Figure 12 depicts the average accuracy of SVM classification on NLTCS, Adult, and TPC-E by Multi-HM algorithm (Multi-HM algorithm is the optimal local differential privacy algorithm commonly used in academia.) [50]. As *f* increases, the classification accuracy shows a downward trend, which also reflects the trade-off of data utility by privacy protection. When *f* is small (f<0.5), the classification accuracy is high and close to the accuracy of the original data. When privacy protection is moderate (i.e., f=0.5), the accuracy of the SVM classification on the synthetic dataset gets higher and closer to that of no privacy protection. This is because the SVM is only influenced by the binary attributes which are generally not sparse, and thus its probability accuracy is high and the correlation is not easy to lose. On the whole, compared with the other local privacy protection method, the high-dimensional data based on local privacy protection in this paper retains good data utility to a certain extent.

## 7. Conclusions

In this paper, we studied better publication of high-dimensional perceptual data under local differential privacy protection in the Crowd-Sensing system. At the beginning, we discussed the existing technology of local privacy protection and high-dimensional data privacy protection, and proposed the local privacy protection of high-dimensional perceptual data based on the Bayes network. In this mechanism, the local differential privacy protection on each user’s data was carried out in the users. Furthermore, after the sensing server receives and aggregates the protected data of each user, we built the Bayes network to illustrate the correlation among attribute dimensions based on the estimation of low-dimension probability distribution and the calculation of mutual information. Besides, in the sequence of the reducing dimensionality and estimating low-dimensional probability distribution based on the constructed Bayes network, a novel dataset was synthesized after sampling the perceptual data under local privacy protection. To verify its effectiveness, we conducted quantities of simulation experiments. Results show that the proposed local privacy protection justified its competence in efficient data publication and privacy protection. Particularly, both multidimensional joint probability distribution query and data classification tasks on synthetic datasets have accuracy close to the original data.

## Figures and Tables

**Figure 1 sensors-20-02516-f001:**
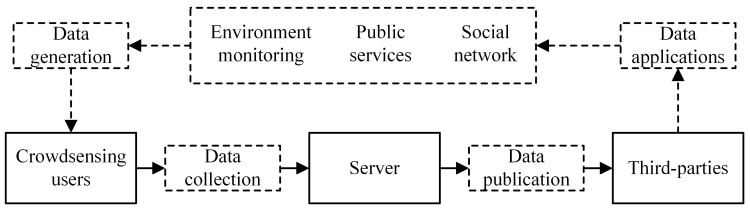
Crowd-Sensing.

**Figure 2 sensors-20-02516-f002:**
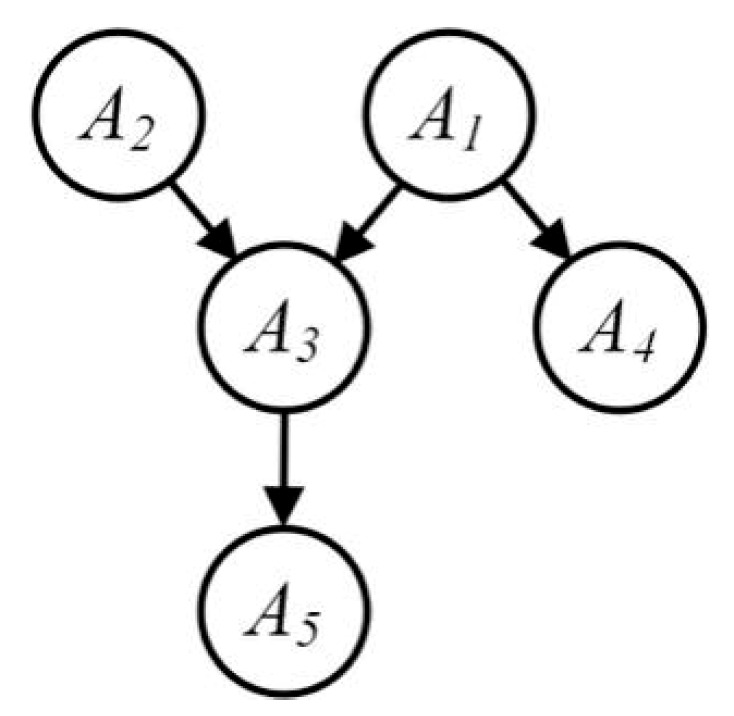
The Bayes network.

**Figure 3 sensors-20-02516-f003:**
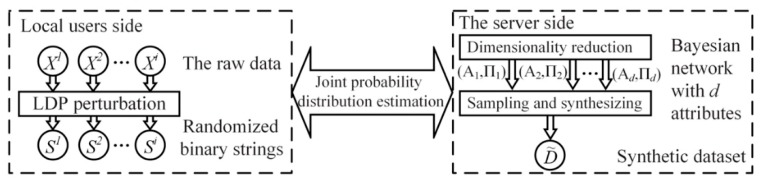
The overview of the local protection on high-dimensional data.

**Figure 4 sensors-20-02516-f004:**
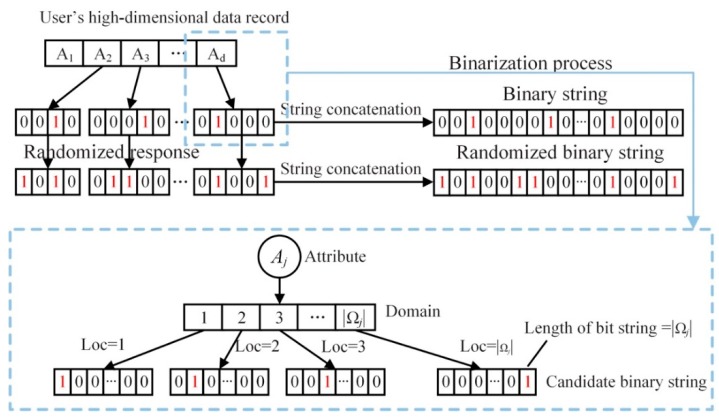
Data binarization.

**Figure 5 sensors-20-02516-f005:**
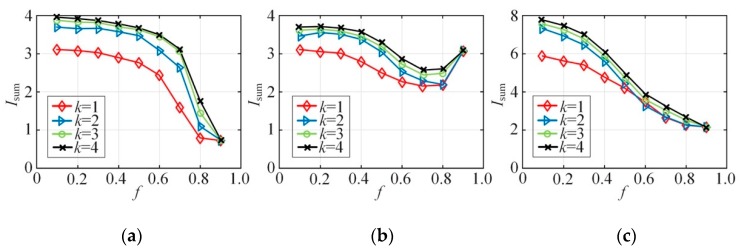
Synthesized dataset $I_{\mathrm{sum}}$ vs. *f*: (**a**) NLTCS, (**b**) Adults, (**c**) TPC-E.

**Figure 6 sensors-20-02516-f006:**
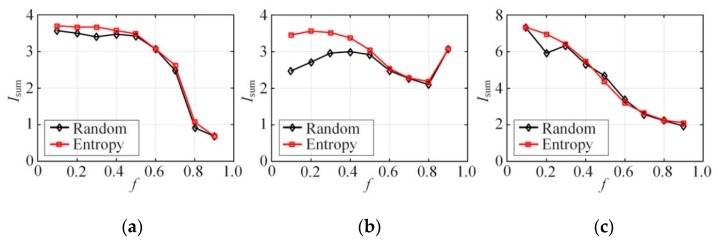
Mutual information $I_{\mathrm{sum}}$ (synthetic dataset) vs. *f* (k = 2): (**a**) NLTCS, (**b**) Adults, (**c**) TPC-E.

**Figure 7 sensors-20-02516-f007:**
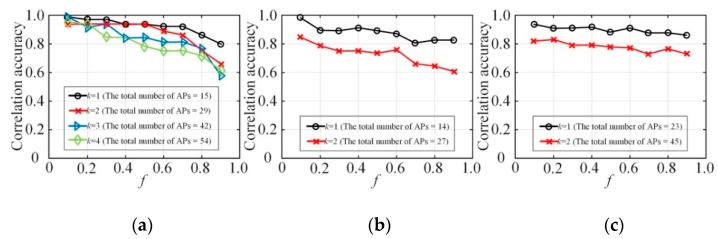
The accuracy of correlation identification: (**a**) NLTCS, (**b**) Adults, (**c**) TPC-E.

**Figure 8 sensors-20-02516-f008:**
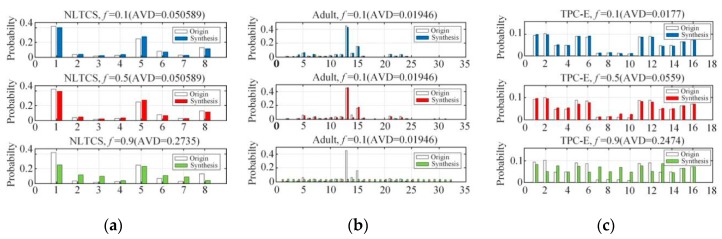
Probability distribution estimation under different ranges |Ω|: (**a**) NLTCS (|Ω|=8), (**b**) Adults (|Ω|=32), (**c**) TPC-E (|Ω|=16).

**Figure 9 sensors-20-02516-f009:**
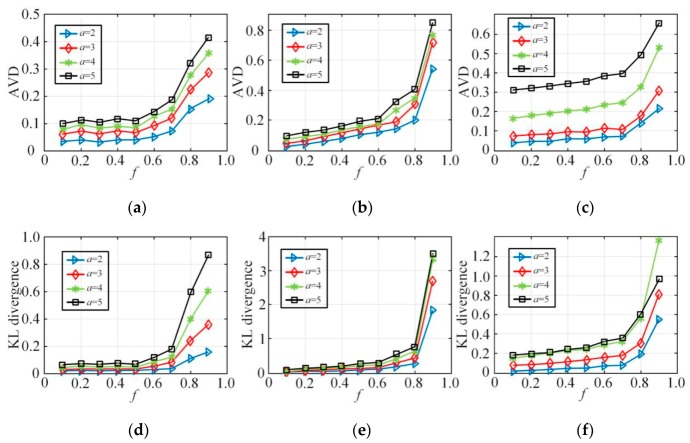
The estimation deviation of a-dimensional joint probability distribution: (**a**) NLTCS (k=2), (**b**) Adults (k=2), (**c**) TPC-E (k=2), (**d**) NLTCS (k=2), (**e**) Adults (k=2), (**f**) TPC-E (k=2).

**Figure 10 sensors-20-02516-f010:**
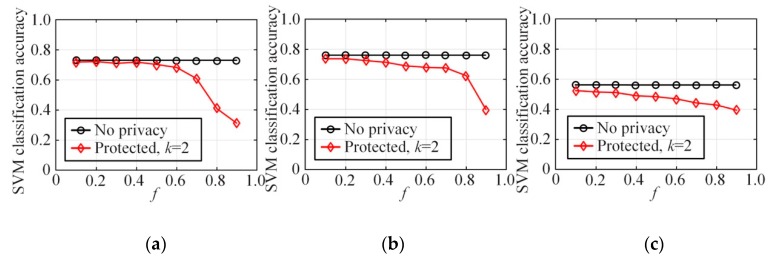
SVM classification accuracy by our method: (**a**) NLTCS, (**b**) Adults, (**c**) TPC-E.

**Figure 11 sensors-20-02516-f011:**
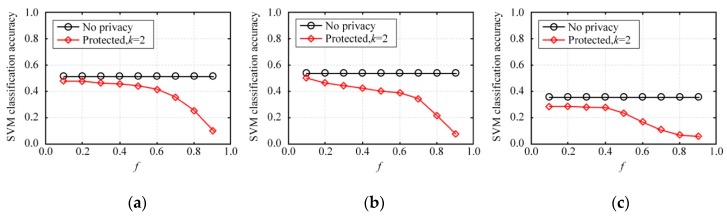
SVM classification accuracy by MeanEST algorithm: (**a**) NLTCS, (**b**) Adults, (**c**) TPC-E.

**Figure 12 sensors-20-02516-f012:**
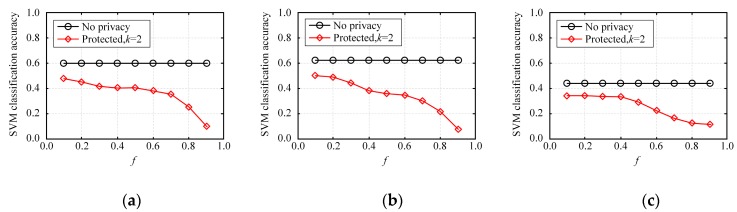
SVM classification accuracy by Multi-HM algorithm: (**a**) NLTCS, (**b**) Adults, (**c**) TPC-E.

**Table 1 sensors-20-02516-t001:** Bayes network parameters.

Attribute	Attribute-Parent Pairs
$A_{1}$	($A_{1},\;\emptyset$)
$A_{2}$	($A_{2},\;\emptyset$)
$A_{3}$	($A_{3},\;\left\{A_{1},\;A_{2}\right\}$)
$A_{4}$	($A_{4},\;\left\{A_{1}\right\}$)
$A_{5}$	($A_{5},\;\left\{A_{3}\right\}$)

**Table 2 sensors-20-02516-t002:** Dataset information description.

Dataset	Data Type	Dataset Size	Dimension	Domain Size (Processed)
NLTCS	Binary	25,174	16	$2^{16}$
Adult	Non-Binary	45,222	15	$\approx 2^{26}$
TPC-E	Non-Binary	40,000	24	$\approx 2^{38}$

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
