# Peer review of "Local Differential Privacy Protection of High-Dimensional Perceptual Data by the Refined Bayes Network"

_sensors, 2020, doi:10.3390/s20092516_

Round 1

Reviewer 1 Report

Talking of Local Differential Privacy Protection is relevant to Sensors readers and this paper presents the technique and the results in a clear way. I would simply suggest the authors to revise the text for typos. Some examples below.

47. As a result, advertising [N]uisance calls, economic loss, and
59. mathematical definition and being flexible in combination. Additionally, [D]ue to b
83. protection for distributed systems, though welcome results have been yielded in so called[]“divide and
134. technology[37] have been proposed in certain literature.
322 self-defined convergence precision[]?
385 [smaller fonts]

Author Response

List of Responses:

Reviewer #1:Talking of Local Differential Privacy Protection is relevant to Sensors readers and this paper presents the technique and the results in a clear way. I would simply suggest the authors to revise the text for typos. Some examples below.

  1. As a result, advertising [N]uisance calls, economic loss, and
  2. mathematical definition and being flexible in combination. Additionally, [D]ue to b
  3. protection for distributed systems, though welcome results have been yielded in so called[]“divide and
  4. technology[37] have been proposed in certain literature. 

322 self-defined convergence precision[]?

385 [smaller fonts]

Dear Editors and Reviewers: 

Thank you for your letter and for the reviewers’ comments concerning our manuscript entitled “Local Differential Privacy Protection of High-dimensional Perceptual Data by the Refined Bayes Network” (ID: sensors-768661). Those comments are all valuable and very helpful for revising and improving our paper, as well as the important guiding significance to our researches. We have studied comments carefully and have made correction which we hope to meet with approval. Revised portion are marked in revisions mode and red in the paper. The main corrections in the paper and the responds to the reviewers’ comments are as flowing:

Responds to the reviewer’s comments: Reviewer #1:

  1. Response to comment:Revise the text for typos: “As a result, advertising [N]uisance calls, economic loss, and”.Response:Considering the Reviewer’s suggestion, we have modified content as “As a result, advertisingharassment, economic loss, and even threats to personal safety will be possible.”
  2. Response to comment:Revise the text for typos: “mathematical definition and being flexible in combination. Additionally, [D]ue to b”.Response:Considering the Reviewer’s suggestion, we have modified content as “DP has emerged as a standard for privacy protection,due to its rigorous mathematical definition and being flexible in combination. Additionally, due to being light-load, differential privacy is particularly suitable for big data processing and scenarios analysis such as Crowd-Sensing data.”
  3. Response to comment:Revise the text for typos: “protection for distributed systems, though welcome results have been yielded in so called[]“divide and”.Response:Considering the Reviewer’s suggestion, we have modified content as “The protections fail to provide effective guarantee on local privacy protection for distributed systems, though welcome results have been yielded in so called “divide and government”.”
  4. Response to comment:Revise the text for typos: “technology[37] have been proposed in certain literature.”.Response:Considering the Reviewer’s suggestion, we have modified content as “The perturbation mechanism based on the compressed input domain [35], the perturbation mechanism based on information distortion [36] and the local differential privacy implementation of randomized response technology [37] have been proposed in certain literature.”
  5. Response to comment:Revise the text for typos: “self-defined convergence precision[]?”.Response:Considering the Reviewer’s suggestion, we have modified content as “According to the self-defined convergence precisionδ, the expected value of the probability distribution is obtained through continuous iteration, which is just what is demanded. The specific process is described below.”
  6. Response to comment:Revise the text for typos: “smaller fonts”.Response:Considering the Reviewer’s suggestion, we have modified the font format of this paper.
  7. Response:Considering the Reviewer’s suggestion, we have modified the format of this paper and references.

Special thanks to you for your good comments.

Finally, according to the suggestion of editors and experts, we have generally checked and adjusted the language of the article, corrected some grammatical errors, and improved the integrity of the article.

We tried our best to improve the manuscript and made some changes in the manuscript.  These changes will not influence the content and framework of the paper. We appreciate for Editors/Reviewers’ warm work earnestly, and hope that the correction will meet with approval.

Once again, thank you very much for your comments and suggestions.

Reviewer 2 Report

This manuscript first proposes an aggregation and publication mechanism for high-dimensional perceptual data of local differential privacy, which finds a pleasant balance between local differential privacy protection and high-dimensional data utility. Furthermore, the authors propose an entropy-inspired estimation for the Bayesian network construction to ensure the correlation between the attributes.

Some of my concerns are as follows:

  1. The English expression of the article is extremely poor, there are massive language errors in this paper, please carefully check it before submitting to the journal. Eg., On Page 1, Line 32, ‘which is’ should be ‘which are’. On Page 2, Line 65, ‘most researches are’ should be ‘most of the research is’. On Page ‘will not be stole’ should be ‘will not be stolen’. On Page 3, Line 90, ‘also find’ should be ‘also finds’, etc.
  2. The authors argue that “The first challenge: Non-local privacy protection. Most of the existing privacy protection researches focus on the processing of the collected data, without considering the privacy exposure risks in the data acquisition process.” However, regardless of whether this ‘challenge’ exists or not, I am not convinced that the authors solved this ‘challenge’ in this manuscript. For details, please refer to the sentence on the Line 161 of the Page 4 of the manuscript, ie., ‘Local privacy protection are carried out after the data records of multiple attribute dimensions are perceived and collected, and then the records are sent to the central server’.
  3. A lot of basic numerical simulations were conducted to quantitatively reveal the effectiveness of the proposed protections. However, there is a lack of comparison with other methods.
    1. References are not detailed enough.·

Author Response

List of Responses:

Reviewer #2:

This manuscript first proposes an aggregation and publication mechanism for high-dimensional perceptual data of local differential privacy, which finds a pleasant balance between local differential privacy protection and high-dimensional data utility. Furthermore, the authors propose an entropy-inspired estimation for the Bayesian network construction to ensure the correlation between the attributes.

Some of my concerns are as follows:

The English expression of the article is extremely poor, there are massive language errors in this paper, please carefully check it before submitting to the journal. Eg., On Page 1, Line 32, ‘which is’ should be ‘which are’. On Page 2, Line 65, ‘most researches are’ should be ‘most of the research is’. On Page ‘will not be stole’ should be ‘will not be stolen’. On Page 3, Line 90, ‘also find’ should be ‘also finds’, etc.

The authors argue that “The first challenge: Non-local privacy protection. Most of the existing privacy protection researches focus on the processing of the collected data, without considering the privacy exposure risks in the data acquisition process.” However, regardless of whether this ‘challenge’ exists or not, I am not convinced that the authors solved this ‘challenge’ in this manuscript. For details, please refer to the sentence on the Line 161 of the Page 4 of the manuscript, ie., ‘Local privacy protection are carried out after the data records of multiple attribute dimensions are perceived and collected, and then the records are sent to the central server’.

A lot of basic numerical simulations were conducted to quantitatively reveal the effectiveness of the proposed protections. However, there is a lack of comparison with other methods.

References are not detailed enough.

Dear Editors and Reviewers: 

Thank you for your letter and for the reviewers’ comments concerning our manuscript entitled “Local Differential Privacy Protection of High-dimensional Perceptual Data by the Refined Bayes Network” (ID: sensors-768661). Those comments are all valuable and very helpful for revising and improving our paper, as well as the important guiding significance to our researches. We have studied comments carefully and have made correction which we hope to meet with approval. Revised portion are marked in revisions mode and red in the paper. The main corrections in the paper and the responds to the reviewers’ comments are as flowing:

Responds to the reviewer’s comments: Reviewer #2

  1. Response to comment:The English expression of the article is extremely poor, there are massive language errors in this paper, please carefully check it before submitting to the journal. Eg., On Page 1, Line 32, ‘which is’ should be ‘which are’. On Page 2, Line 65, ‘most researches are’ should be ‘most of the research is’. On Page ‘will not be stole’ should be ‘will not be stolen’. On Page 3, Line 90, ‘also find’ should be ‘also finds’, etc. Response:Considering the Reviewer’s suggestion, we have modified content all above mentioned.We have further modified and optimized the full-text syntax and language.
  2. Response to comment:The authorsargue that “The first challenge: Non-local privacy protection. Most of the existing privacy protection researches focus on the processing of the collected data, without considering the privacy exposure risks in the data acquisition process.” However, regardless of whether this ‘challenge’ exists or not, I am not convinced that the authors solved this ‘challenge’ in this manuscript. For details, please refer to the sentence on the Line 161 of the Page 4 of the manuscript, ie., ‘Local privacy protection are carried out after the data records of multiple attribute dimensions are perceived and collected, and then the records are sent to the central server’.Response:Considering the Reviewer’s suggestion, we have modified and add content as “The Crowd-Sensing system in this paper consists of a large number of sensing users which are connected to a central server. Local privacy protection is carried out after the records of multiple attribute dimensions are perceived and collected, and then the records are sent to the central server. The server collects all locally protected data, estimates and analyzes the statistical distribution of high-dimensional data, and then synthesizes a novel data set of approximate distribution for third-party users for public query and mining. It must be admitted that the method proposed in this paper does not completely solve the challenge of non-local privacy protection, but it also explores the solution to this challenge to some extent. Here, we mainly focus on data privacy, so we do not consider specific network models.”
  3. Response to comment: A lot of basic numerical simulations were conducted to quantitatively reveal the effectiveness of the proposed protections. However, there is a lack of comparison with other methods.Response:Considering the Reviewer’s suggestion, we have modified and add content as “Figure 10 depicts the average accuracy of SVM classification on NLTCS, Adult, and TPC-ï¹£E by our method. Figure 11 depicts the average accuracy of SVM classification on NLTCS, Adult, and TPC-E by MeanEST algorithm [49]. Figure 12 depicts the average accuracy of SVM classification on NLTCS, Adult, and TPC-E by Multi-HM algorithm [50]. As f increases, the classification accuracy shows a downward trend, which also reflects the trade-off of data utility by privacy protection. When f is small (f<0.5), the classification accuracy is high and close to the accuracy of the original data. When privacy protection is moderate (i.e., f=0.5), the accuracy of the SVM classification on the synthetic dataset gets higher and closer to that of no privacy protection. This is because the SVM is only influenced by the binary attributes which are generally not sparse, and thus its probability accuracy is high and the correlation is not easy to lose. On the whole, compare with other local privacy protection method, the high-dimensional data based on local privacy protection in this paper retains good data utility to a certain extent.” And we also add the figure 11 and figure 12.
  4. Response to comment: References are not detailed enough.Response:Considering the Reviewer’s suggestion, we have modified the reference list as detail as possible to adapt the format for “sensors” journal.
  5. Response:Considering the Reviewer’s suggestion, we have modified the font format of this paper.
  6. Response:Considering the Reviewer’s suggestion, we have modified the format of this paper and references.

Special thanks to you for your good comments.

Finally, according to the suggestion of editors and experts, we have generally checked and adjusted the language of the article, corrected some grammatical errors, and improved the integrity of the article.

We tried our best to improve the manuscript and made some changes in the manuscript.  These changes will not influence the content and framework of the paper. We appreciate for Editors/Reviewers’ warm work earnestly, and hope that the correction will meet with approval.

Once again, thank you very much for your comments and suggestions.

Reviewer 3 Report

Summary:
In the paper "Local Differential Privacy Protection of High-dimensional Perceptual Data by the Refined Bayes Network", an approach
is investigated to protect "local differential privacy" in mobile crowdsensing scenarios. Bayesian networks, in turn, are the major
pillar of the technical approach of the authors. The authors describe related work, background information, their technical contribution, and
finally, experimental results. The authors conclude that through "experiments, the proposed local privacy protection has been justified its
competence ...".

Points in favor:
- The topic fits to the scope of the journal
- The authors deal with a topical subject
- The authors discuss related works
- The authors show experimental results

Points against the paper:
- In general, the paper has very many language issues and is very difficult to read,
e.g.,
(1) What is perceptual data, the term is not explained, it is easy to assume what is meant, however, the term is not precise
(2) Many sentences can be mentioned that are difficult to read
(3) Language is not consistent, sometimes Bayesian network, sometimes Bayes Network
(4) As an example, after reading the abstract, it is not clear what the mission of the paper is
... in addition: many more language issues are a major problem
- Mobile crowdsensing is never defined, and not explained, which flavor is addressed here: opportunistic, participatory, ....
- A serious flaw is that the reader shall know many things that are never explained
- Many statements are very confusing: For example, in the abstract: "Thus ..."
-> haveing the sentences before this one in mind, most things come out of the blue
- Abstract and Conclusions does not fit semantically
- Figure 1 could have a better quality
- Structure of the paper is not given

-> As the most negative point, the paper needs language-wise a major, major revision, therefore my recommendation is to revise the paper very substantially in the next round, otherwise I would vote for a reject in the next round

Author Response

List of Responses:

Reviewer #3:

Summary:

In the paper "Local Differential Privacy Protection of High-dimensional Perceptual Data by the Refined Bayes Network", an approach

is investigated to protect "local differential privacy" in mobile crowdsensing scenarios. Bayesian networks, in turn, are the major

pillar of the technical approach of the authors. The authors describe related work, background information, their technical contribution, and

finally, experimental results. The authors conclude that through "experiments, the proposed local privacy protection has been justified its

competence ...".

Points in favor:

- The topic fits to the scope of the journal

- The authors deal with a topical subject

- The authors discuss related works

- The authors show experimental results

Points against the paper:

- In general, the paper has very many language issues and is very difficult to read,

e.g.,

(1) What is perceptual data, the term is not explained, it is easy to assume what is meant, however, the term is not precise

(2) Many sentences can be mentioned that are difficult to read

(3) Language is not consistent, sometimes Bayesian network, sometimes Bayes Network

(4) As an example, after reading the abstract, it is not clear what the mission of the paper is

... in addition: many more language issues are a major problem

- Mobile crowdsensing is never defined, and not explained, which flavor is addressed here: opportunistic, participatory, ....

- A serious flaw is that the reader shall know many things that are never explained

- Many statements are very confusing: For example, in the abstract: "Thus ..."

-> haveing the sentences before this one in mind, most things come out of the blue

- Abstract and Conclusions does not fit semantically

- Figure 1 could have a better quality

- Structure of the paper is not given

-> As the most negative point, the paper needs language-wise a major, major revision, therefore my recommendation is to revise the paper very substantially in the next round, otherwise I would vote for a reject in the next round

Dear Editors and Reviewers: 

Thank you for your letter and for the reviewers’ comments concerning our manuscript entitled “Local Differential Privacy Protection of High-dimensional Perceptual Data by the Refined Bayes Network” (ID: sensors-768661). Those comments are all valuable and very helpful for revising and improving our paper, as well as the important guiding significance to our researches. We have studied comments carefully and have made correction which we hope to meet with approval. Revised portion are marked in revisions mode and red in the paper. The main corrections in the paper and the responds to the reviewers’ comments are as flowing:

Responds to the reviewer’s comments: Reviewer #3

  1. Response to comment: In general, the paper has very many language issues and is very difficult to read. Response:Considering the Reviewer’s suggestion, we have systematic modified the language in this paper.
  2. Response to comment: What is perceptual data, the term is not explained, it is easy to assume what is meant, however, the term is not precise.Response:Considering the Reviewer’s suggestion, we have modified and add content as “The booming in equipment manufacturing, communication technology, data processing, algorithm, together with the emergence of Internet of Things (IOT), gives rise to the Crowd-Sensing [1,2], a key access to the formation of information value service from the physical world. As shown in Figure 1, various smart devices, which are portable in large space, can realize the perception and digitization of the physical world across time and space. Consequently, large-scale data are acquired for the Crowd-Sensing system [3]. The data are then published to third-party users via sensing servers to perform various analysis, mining and machine learning, ended with providing accurate feedback and decision-making guidance for social production and life [4]. In addition to its large scale, Crowd-Sensing data obtained by immense heterogeneous sensing devices boast the attributes of multi-dimension or even high-dimensional characteristics in many cases, and therefore, mining the correlation among the attribute dimensions is vital to the value of Crowd-Sensing data. For example, the correlation analysis of physical features in a patient's health record helps in the prediction and discovery of potential disease [5], and the correlation analysis of shopping and browsing behaviors of mobile phone users facilitates the personalized recommendation system [6]. Mobile crowd-Sensing perception is to take the user's smart mobile device as the basic perception unit, carry out conscious or unconscious collaboration through the mobile internet, realize the distribution of perception and perception data collection, so as to effectively complete the large-scale perception tasks in the fields of urban traffic, society and environment. With the development of sensors, the modes of Crowd-Sensing perception data tend to be diversified. In addition to sensor data in traditional digital form, more and more Crowd-Sensing perception data are presented in various forms such as sound, image and text.”
  3. Response to comment: Mobile crowdsensing is never defined, and not explained, which flavor is addressed here: opportunistic, participatory, ....Response:Considering the Reviewer’s suggestion, we have modified and add content as “Mobile crowd-Sensing perception is to take the user's smart mobile device as the basic perception unit, carry out conscious or unconscious collaboration through the mobile internet, realize the distribution of perception and perception data collection, so as to effectively complete the large-scale perception tasks in the fields of urban traffic, society and environment.”
  4. Response to comment:Language is not consistent, sometimes Bayesian network, sometimes Bayes Network.Response:Considering the Reviewer’s suggestion, we have agreed on two statements to “Bayes”.
  5. Response to comment:Abstract and Conclusions does not fit semantically.Response:Considering the Reviewer’s suggestion, we have modified the abstract as “Although the Crowd-Sensing perception system brings great data value to people through the release and analysis of high-dimensional perception data, it causes great hidden danger to the privacy of participants in the meantime. Currently, various privacy protection methods based on differential privacy have been proposed, but most of them cannot simultaneously solve the complex attribute association problem between high-dimensional perception data and the privacy threat problems from untrustworthy servers. To address this problem, we put forward a local privacy protection based on Bayes network for high-dimensional perceptual data in this paper. This mechanism realizes the local data protection of the users at the very beginning, eliminates the possibility of other parties directly accessing the user's original data, and fundamentally protects the user's data privacy. During this process, after receiving the data of the user's local privacy protection, the perception server recognizes the dimensional correlation of the high-dimensional data based on the Bayes network, divides the high-dimensional data attribute set into multiple relatively independent low-dimensional attribute sets, and then sequentially synthesizes the new data set. It can effectively retain the attribute dimension correlation of the original perception data, and ensure that the synthetic data set and the original data set have as similar statistical characteristics as possible. To verify its effectiveness, we conduct a multitude of simulation experiments. Results have shown that the synthetic data of this mechanism under the effective local privacy protection has relatively high data utility.” Considering the Reviewer’s suggestion, we have modified the conclusion as “In this paper, we study better publication of high-dimensional perceptual data under local differential privacy protection in the Crowd-Sensing system. At the beginning, we discuss the existing technology of local privacy protection and high-dimensional data privacy protection are discussed at the very beginning, and propose the local privacy protection of high-dimensional perceptual data based on the Bayes network. In this mechanism, the local differential privacy protection on each user's data is carried out in the users. Furthermore, after the sensing server receives and aggregates the protected data of each user, we build the Bayes network to illustrate the correlation among attribute dimensions based on the estimation of low-dimension probability distribution and the calculation of mutual information. Besides, in the sequence of the reducing dimensionality and estimating low-dimensional probability distribution based on the constructed Bayes network, a novel data set is synthesized after sampling the perceptual data under local privacy protection. To verify its effectiveness, we conduct quantities of simulation experiments. Results show that the proposed local privacy protection has been justified its competence in efficient data publication and privacy protection. Particularly, both multi-dimensional joint probability distribution query and data classification tasks on synthetic data sets have accuracy close to the original data.”
  6. Response to comment: Figure 1 could have a better quality.Response:Considering the Reviewer’s suggestion, we have modified the Figure 1.
  7. Response to comment:Structure of the paper is not given. Response:Considering the Reviewer’s suggestion, we have modified and add content as “The remainder of this paper is organized as follows. Section 2 describes related work. Section 3 introduce system model. In Section 4, we introduce some necessary basic knowledge. Section 5 describes the protection algorithm of local differential privacy on high-dimensional perceptual data. Experimental evaluation results are provided in Section 6. Finally, in Section 7, we conclude this paper.”
  8. Response:Considering the Reviewer’s suggestion, we havemade subversive changes to the language expression of the full text to ensure smooth logic and improve the readability of the paper.
  9. Response:Considering the Reviewer’s suggestion,we have modified some sentences which difficult to read.
  10. Response:Considering the Reviewer’s suggestion, we have modified the font format of this paper.
  11. Response:Considering the Reviewer’s suggestion, we have modified the format of this paper and references.

Special thanks to you for your good comments.

Finally, according to the suggestion of editors and experts, we have generally checked and adjusted the language of the article, corrected some grammatical errors, and improved the integrity of the article.

We tried our best to improve the manuscript and made some changes in the manuscript.  These changes will not influence the content and framework of the paper. We appreciate for Editors/Reviewers’ warm work earnestly, and hope that the correction will meet with approval.

Once again, thank you very much for your comments and suggestions.

Round 2

Reviewer 2 Report

The authors have made substantial efforts to revise the manuscript according to my comments, and the quality of the revised revision has improved a lot. 

Reviewer 3 Report

The changes are sufficient for an acceptance.